# Patient-Reported Outcome Measures for Evaluating Body Awareness: A Systematic Review Using the COSMIN Methodology

**DOI:** 10.3390/healthcare13243270

**Published:** 2025-12-12

**Authors:** Cristina Bravo, Manuel Trinidad-Fernández, David Barranco-i-Reixachs, Sandy Arias-Matiz, Pedro Malagon-Santos, Daniel Catalán-Matamoros

**Affiliations:** 1Departament d’Infermeria i Fisioteràpia, Universitat de Lleida, Carrer de Montserrat Roig, 2, 25198 Lleida, Spain; cristina.bravo@udl.cat (C.B.); david.barranco@udl.cat (D.B.-i.-R.); 2Grup d’Estudis Societat, Salut, Educació i Cultura (GESEC), Universitat de Lleida, Carrer Jaume II, 73, 25001 Lleida, Spain; 3Health Care Research Group (GRECS), Lleida Institute for Biomedical Research Dr. Pifarré Foundation, Av. Alcalde Rovira Roure, 80, 25198 Lleida, Spain; 4Departamento de Movimiento Corporal Humano, Facultad de Medicina, Universidad Nacional de Colombia—Sede Bogotá, Carrera 30 No. 45-03, Edificio 224, Bogotá D.C. 111321, Colombia; 5Research Group CTS 451 “Health Sciences”, University of Almeria, 04120 Almeria, Spain; dcatalan@ual.es; 6MediaLab, University Carlos III of Madrid, 28903 Madrid, Spain

**Keywords:** body awareness, interoceptive awareness, embodiment, systematic review, PROM, reliability, validity

## Abstract

**Objective**: Body awareness is the conscious, subjective multimodal integration of body-related sensitivity from bodily signals—detecting states and subtle reactions to internal and environmental conditions—modifiable by attention, interpretation, appraisal, beliefs, memories, conditioning, attitudes, and affect. The aim of our study is to identify patient-reported outcome measures (PROMs) of BA and evaluate their psychometric properties and cross-cultural adaptation processes. **Literature Survey**: We searched PubMed, Scopus, and PsycINFO; the last search was conducted on 1 July 2025. **Methodology:** We included studies that psychometrically evaluated PROMs regarding BA in the general adult population and their translations into other languages, with no time-range restrictions. Study selection was performed independently by two reviewers in a blind manner. Evaluation followed COSMIN guidance for systematic reviews of PROMs: (1) risk of bias assessment, (2) application of quality criteria for measurement properties, and (3) GRADE rating of the certainty of evidence. **Synthesis:** We identified 12 BA questionnaires and more than 30 cross-cultural adaptations, from a total of 50 studies. In summary, the Revised Body Awareness Rating Questionnaire and the Multidimensional Assessment of Interoceptive Awareness (MAIA 1 and 2) showed good results for structural validity and internal consistency, which were the most frequently assessed psychometric properties. In contrast, construct validity was highly variable, and the findings on reliability were far from optimal. MAIA-2 was one of the most studied and showed stronger evidence and better pooled results (4 out of 5 properties) than other instruments. **Conclusions:** The psychometric quality of BA PROMs varies widely, reflecting challenges in operationalizing the construct of body awareness and related domains. While MAIA-2 currently presents the most acceptable—though still imperfect—evidence, further high-quality studies are needed to strengthen their measurement properties and clarify construct coverage.

## 1. Introduction

Recently, interest in body awareness (BA) and its implications for health has grown. The underlying mechanisms of how our organism reacts to internal and external stimuli, and how we perceive the body and interpret reality, are important topics for explaining the latest evidence on body awareness interventions. Despite minor differences in emphasis, there is general agreement that body awareness (BA) refers to the subjective and multimodal integration of bodily signals. Most definitions highlight a dynamic interaction between sensing internal states and the cognitive–affective processes that follow—such as attention, interpretation, and evaluation—which together shape one’s subjective experience of the body [1,2,3,4,5]. Thus, BA can be understood as the conscious, subjective multimodal integration of body-related sensitivity from bodily signals—detecting states and subtle reactions to internal and environmental conditions— modifiable by attention, interpretation, appraisal, beliefs, memories, conditioning, attitudes, and affect. This conceptual complexity has led to the development of numerous patient-reported outcome measures (PROMs), each capturing different aspects of this multifaceted construct.

The disruption of BA can also be found in many mental health disorders such as anxiety, depression, or eating disorders [6,7,8] and other conditions such as pain severity, catastrophizing, disability, or cognitive impairment [9,10,11]. Consequently, instruments for assessing BA have been developed and validated in the last few years. Self-reports provide indispensable information about internal experiences in naturalistic settings and provide a crucial complement to laboratory-based measures, physiological monitoring, and qualification of neural processes [12].

Nowadays, several self-reported instruments or patient-reported outcome measures (PROMs) exist that assess BA, but there is a lack of a gold-standard instrument. It is essential to analyse if they are adequate, reliable, multidimensional, and standardized to assess all the complex aspects of BA and to find a reference scale. We hypothesized that tools specifically designed to assess multidimensional aspects of BA, such as MAIA or MAIA-2, would demonstrate stronger psychometric performance compared to more general or unidimensional measures. Therefore, this systematic review has two primary aims: first, identifying instruments that could assess BA and describing their psychometric properties; second, finding which of the huge variety of self-reported instruments is the most rigorous by making a comparison and a critical revision of all of them.

### 1.1. Methodology and Design

This study is a systematic review of PROMs on body awareness in the adult population (healthy and unhealthy) developed based on the Preferred Reporting Items for Systematic Reviews and Meta-Analyses (PRISMA) guidelines [13] and COSMIN guidelines for systematic reviews of PROMs recommendations for performing a systematic review of measurement properties [14]. It was registered previously on the PROSPERO database at the beginning of the literature search (CRD42023471289).

### 1.2. Search Strategy

The electronic databases used in the search were PubMed, Scopus, and PsycINFO. The investigation was restricted to human studies conducted in the English and Spanish languages only. The search had no restrictions on publication years; no initial publication date was applied. The final search date, which also served as the end date for the search period, was 1 July 2025. More complete information about search strategy is provided in Appendix A. The selected terminology was based on Medical Subject Heading (MeSH), Scopus, and PsycINFO Thesaurus. It combined two broad concepts: (i) body awareness; (ii) PROMs, including questionnaires and scales self-completed by the patient.

### 1.3. Study Selection

Observational validation studies were selected meeting the framework suggested by COSMIN guidelines: the construct was body awareness according to the current definition [2,4], the target population was the general adult population, the type of instrument was only PROMs, and all the measurement proprieties were searched. The measurement properties according to the COSMIN guidelines are content validity, structural validity, internal consistency, reliability, hypothesis testing for construct validity, criterion validity, measurement error, cross-cultural validity, and responsiveness. Definitions are detailed in Appendix A. The Rayyan app was used for the article selection process [15].

### 1.4. Study Inclusion

First, the search was cleared of duplicate articles. The titles and abstracts were assessed separately and blinded for eligibility by two reviewers. Studies that fully satisfied the following criteria during the title and abstract screening were excluded in the review: (i) not PROM psychometric analysis; (ii) not body awareness; (iii) not primary documents. The reading of the full text was the following step, when the exclusion criteria were (i) not PROMs and (ii) not body awareness. The evaluators thoroughly analyzed the construct and the scale items to ensure that they accurately reflected the definition of BA. When an agreement could not be reached on a document, a third reviewer provided arbitration. To assess the level of agreement between reviewers during the full-text screening phase, we calculated both the observed agreement and Cohen’s kappa, which adjusts for the agreement expected by chance. Interpretation of κ followed the conventional Landis and Koch benchmarks [16]: values < 0.00 indicate poor agreement, 0.00–0.20 slight agreement, 0.21–0.40 fair agreement, 0.41–0.60 moderate agreement, 0.61–0.80 substantial agreement, and 0.81–1.00 almost perfect agreement.

### 1.5. Data Analysis

The assessment of every measurement attribute incorporated in this research adhered to the COSMIN methodology for systematic reviews of PROMs [14,17]. The evaluation was divided in three steps.
1.Evaluation of the methodological quality of the included studies using the COSMIN risk of bias checklist: During this phase, each measurement property in the chosen studies was evaluated. The risk of bias for each study was assigned a rating on a four-point scale, classifying it as either very good, adequate, doubtful, or inadequate quality. The lowest rating among the items was selected for each measurement property [17,18].2.Quality criteria for good measurement properties: The findings from studies on all measurement properties were summarized qualitatively, and each result was assigned a rating of sufficient (+), insufficient (–), or indeterminate (?) [17,18]. More information is available in Appendix A. Due to the great number of subscales in some PROMs, the symbol ± will be used when, in the analysis of subscales, a value from one subscale does not meet the cutoff point while the rest do. Additionally, for the property “Hypotheses testing for construct validity,” a cutoff point of r = 0.30 was used (positive for convergent validity and negative for divergent validity) to ensure that the measurement is relevant and confirms the hypothesis [19].3.Summary of the evidence and GRADE quality of the evidence: The consistency of the measurement properties results was validated by scrutinizing all available studies. The outcomes were aggregated and assessed against the modified GRADE criteria, categorizing them as high, moderate, low, or very low according to the COSMIN manual. The GRADE approach employs five dimensions (risk of bias, inconsistency, indirectness, and imprecision) to determine and score the quality of evidence [17,18].

Given the potential to encounter translations or modifications in alternative languages, the translation process was also analyzed following the Beaton guidelines [20]. Five steps were checked for the translation process: translation to the new language, synthesis, back-translation, debriefing, and questionnaire test of the prefinal version with the related sample. A point was awarded for each phase if it was explicitly mentioned in the study and had been carried out correctly according to the Beaton guidelines.

### 1.6. Data Extraction and Data Synthesis

The extracted information included the patient-reported outcome measure (PROM), name of the PROM, characteristics of the PROM, country and language of study, sample size, type of data collection, average age, gender, target population, translation process (if applicable), measurement properties evaluated, and psychometric results. Additionally, the follow-up time and the type of intervention were noted if the reliability and the responsiveness were analyzed, respectively.

Two researchers independently conducted this process to ensure comprehensive data retrieval and, subsequently, the datasets were combined. The synthesized information was organized into three tables: study characteristics, quality of the psychometric outcomes, and pooled results with the quality of evidence.

## 2. Results

### 2.1. Systematic Literature Search

A total of 5250 articles were found in the three databases used. An additional 6 articles were found from other sources, bringing the total to 5256 articles that began the selection process. After removing duplicate articles, 3455 articles were screened by reading the title and abstract. In the next phase, 66 articles were analyzed by reading the full text, resulting in 50 articles being included in the review. During the full-text screening phase, the observed agreement between reviewers was 0.89, and Cohen’s kappa was 0.70, indicating a substantial level of agreement after accounting for chance. Among these articles, 12 different BA questionnaires were identified, along with more than 30 translations of these questionnaires. The full process is specified in Figure 1.

### 2.2. Descriptions of the Included PROMs

The different 12 BA PROMs were found after the selection process. The full description of the questionnaires is provided in Table 1. The total number of items ranges from 12 to 46, with the ABC based on a body chart with 51 regions [21]. Six of the ten questionnaires also include subscales [12,22,23,24,25,26]. Regarding the languages, most of the original questionnaires were in English (58%), and some of the scales were translated into other 15 different languages. Concerning the possible responses, the questionnaires used a Likert scale with 5 to 7 response options.

Some of these questionnaires are shortened versions or adaptations for specific populations. For example, BPQ-VSF and BARQ-R are even more reduced versions of BPQ-SF and BARQ, respectively [12,28], MAIA-2 is an optimized version with more items by the original authors of the first MAIA [22,26], and Brief MAIA-2 is a shortened version proposed by a different team of authors than the original creators [25]. PBE-QAG is an adaptation of the PABE for older adults [23,30]. Since these are derived scales, they have a different number of items and different objectives, which is why their psychometric properties differ, and they were analyzed as distinct scales.

### 2.3. Characteristics of the Studies

The properties measured in the questionnaires included in the systematic review were content validity, structural validity, internal consistency, reliability, hypothesis testing for construct validity (especially convergent, divergent, and known validity), criterion validity, responsiveness, and measurement error.

The participants in these studies were very diverse. Most were healthy adults or university students, but there were also older adults and patients with conditions such as rheumatoid arthritis, chronic pain, non-specific low back pain, or eating disorders, for example. The average age ranged from 19.6 to 70.3 years. The questionnaires were collected using two main methods: paper (in person or by mail) or computerized (via online surveys or using an electronic terminal in person).

The characteristics of the included studies are in Table 2.

### 2.4. Measurement Properties and Quality of Evidence of PROMs

A comprehensive analysis of the included studies was conducted to extract all relevant information on the psychometric properties of the questionnaires and to present the most significant results according to the COSMIN guidelines. Appendix A displays the evaluation of methodological quality, the quality of the measured properties for each version of the included questionnaires individually, and, in the case of translated scales, the cross-cultural translation process. This translation process is detailed further in Appendix A. There were several studies that used translated versions using the process performed in another previous study. All quantified results for each property of each questionnaire can be found in Appendix A.

Additionally, Table 3 provides a combined overview of the twelve questionnaires: the number of properties analyzed, a summary of the methodological quality and the quality of the measured properties, and the quality of evidence assessment according to the GRADE approach in PROMs. In the case of content validity, the ROB analysis of all of the original questionnaires received the designation of Doubtful. The weakest aspects of this property were that it was not clear whether professionals participated in the development of the PROM and that it did not involve at least two researchers in the analysis.

Next, the most important details of the twelve questionnaires analyzed through the included studies will be presented.

### 2.5. ABC

The ABC scale was only analyzed in its original version in a single study; therefore, a summary analysis could not be performed. Despite having good internal consistency, convergent validity, and reliability, the structural validity could not be adequately tested without confirmatory structural analysis.

### 2.6. BARQ and BARQ-R

Five properties were analyzed for BARQ (in three studies). Structural validity was assessed once, but the results were not satisfactory according to COSMIN guidelines. Because of this unsatisfactory structural validity, unidimensionality was not demonstrated, which affected the internal consistency analysis. The reliability analysis showed a very low quality of evidence, and the construct validity results were inconsistent. Only one translation into Turkish was carried out, completing 4 out of 5 steps of the translation process.

The revised version of the original questionnaire (BARQ-R) produced better results and showed a more clearly defined instrument due to the reduction in the number of items. Structural validity, internal consistency, and reliability all demonstrated good results, with a high level of evidence and a low risk of bias. However, only two studies have tested the new version. The authors of the original questionnaire also produced a translation into English, but no details were provided about the cross-cultural adaptation process.

### 2.7. BAQ

Five properties were analyzed for this questionnaire and the translations. Reliability showed good results in terms of pooled results. On the other hand, structural validity and convergent validity were not confirmed due to insufficient results. In addition, most of the evidence-quality assessments received a low rating. Known validity and responsiveness showed good results, but they were measured in only 1 or 2 studies.

Regarding cross-cultural translations, 2 out of 6 questionnaires were successfully translated with 4 out of 5 steps of the process completed.

### 2.8. BPQ-SF and BPQ-VSF

Four properties were analyzed for BPQ-SF in its original version, conducted simultaneously in two languages (English and Spanish) within the same study, along with three translations. The methodological quality was good with a high level of evidence for internal consistency and construct validity, but somewhat lower with a moderate and low level of evidence for structural validity and reliability, respectively. The quality of the evidence was sufficient or inconsistent.

Regarding the translations, the Italian and Persian versions performed the process satisfactorily (4 out of 5 steps), while the Chinese version provided very little information about the process (1 out of 5 steps).

BPQ-VSF is a shortened version of the previous one and was analyzed in the same studies as the earlier questionnaires. It obtained similar results to its predecessor. Structural validity was not tested but one study assessed criterion validity with the predecessor, and the correlation was quite high (r = 0.94).

### 2.9. MAIA, MAIA-2, and Brief MAIA-2

Five properties were analyzed in the MAIA questionnaire. It is the most translated and studied questionnaire in the systematic review. The structural validity varied greatly in terms of methodological quality across the different versions, with inconsistent measurements but moderate evidence quality. Internal consistency, reliability, and construct validity were inconsistent or insufficient because some subscales produced better results than others. For example, the subscales “not worrying” and “not distracting” were the most inconsistent across studies. Some articles even found better models without any subscale from the original. This inconsistency also extended to divergent validity and responsiveness, but these properties were not analyzed as extensively. The methodological quality was good for these properties, except for reliability, which was classified as very low. Finally, the translation process was better detailed in four articles (with more than four steps completed), but the majority of translations did not meet this level of detail.

Four properties were analyzed in this second version of the MAIA questionnaire (MAIA-2), which was developed by the same authors as the original. The results were similar to those reported for the original MAIA, except for improvements in structural validity, internal consistency, and convergent validity, which reached higher levels of measurement quality, as well as better reliability results with a low quality of evidence. Some subscales still showed insufficient results, but to a lesser extent than in the original version. Additionally, the quality of evidence ranged from moderate to high for the properties analyzed, while reliability remained low overall. More than half of the translated questionnaires obtained more than 3 out of 5 fulfilled steps.

Two properties were analyzed in this shortened version of MAIA-2 translated by Polish authors. There are no other adaptations into other languages. The results were sufficient, except for some poorer outcomes in certain subscales, but the methodological quality of the structural validity was doubtful. The translation process was not well-detailed, as only 2 out of the 5 steps were mentioned.

### 2.10. PABE and PBE-QAG

Only the original version of PABE was found. The methodological quality of the four properties measured ranged from very good to doubtful. The results were sufficient in internal consistency and convergent validity but insufficient and indeterminate in structural validity, reliability, and divergent validity.

Two properties were analyzed in the original version of PBE-QAG and a translation in this adaptation of the previous questionnaire. Only two studies evaluated structural validity, and the pooled result was insufficient, supported by low-quality evidence.

### 2.11. SBC

Five properties were measured in the studies that analyzed this questionnaire and the translations. Two studies provide most of the properties analyzed in different languages, but there is no information about the cross-cultural translation process. The other translations that detail the process do not show satisfactory adherence to the guidelines (2 or 3 out of 5 steps). The quality of the measurement was sufficient and inconsistent due to differences between subscales. The quality of evidence ranged from very low to low, except for internal consistency.

## 3. Discussion

### 3.1. Systematic Literature Search

The evaluation of patient-reported outcome measures (PROMs) for complex constructs, such as body awareness, presents unique challenges. In the absence of a clear and multidimensional consensus view on the construct, substantial inconsistencies and heterogeneity in conceptual interpretations remain [69,70], suggesting that many of the questionnaires may not in fact be assessing precisely what they intend to measure. Nevertheless, BA is still a concept widely used in patients with chronic pain such as fibromyalgia [71] or musculoskeletal pain [72] and to measure the detrimental impact on body image that influences the intensity of reported pain [73]. These challenges are not unique to body awareness but are also observed in other multidimensional constructs, such as quality of life and patient satisfaction [74,75], where diverse and overlapping measures often lack consistency and clarity. In addition, it can be observed that these scales have been assessed primarily in students or relatively young individuals. Professionals from various relevant disciplines, such as psychology, medicine, or physiotherapy, were not included in all of the creation processes. These facts could be the main reasons for the doubtful rating in content validity analysis.

In this discussion, we aim to contextualize our findings within the broader literature, explore the implications for clinical and research applications, and propose recommendations to enhance the development and validation of PROMs for complex constructs. In short, no BA PROM was found to have excellent results or to be considered a reference instrument, unlike what occurs with instruments that assess other aspects, such as sleep in fibromyalgia [76]. Our findings highlight several methodological limitations in the validation processes of these instruments, which impact the overall quality.

### 3.2. ABC

Regarding the ABC scale, we only have a psychometric analysis from the original authors and there is a lack of information to determine whether the scale is recommendable or not. Utilizing bodily drawing representations may serve as a more effective instrument for evaluating complex cases than the conventional questionnaire-based method, but more studies are needed. In contrast to another scale utilizing bodily drawing representations for the assessment of pain, it had a reliability more robust with ICC values reported as highly positive [77]. Overall, the ABC scale offers an innovative approach to assessing body awareness through bodily drawing representations, but the evidence base remains very limited.

### 3.3. BARQ and BARQ-R

BARQ showed psychometric improvements in the revised version. The review of the questionnaire granted BAQ-R better structural validity, although convergent validity has not yet been assessed, a property for which the results were inconsistent in the previous version. Although only two studies have evaluated BARQ-R, it seems that the validation process and the properties obtained are apparently excellent. Further studies on the validated tool are needed, as it appears it should replace the previous version, as well as more cross-cultural adaptations into other languages with their corresponding detailed processes, something that is not specified in the English version.

### 3.4. BAQ

Despite demonstrating a better reliability, BAQ exhibits a recurring pattern of unsatisfactory structural validity across multiple studies (only 2 of the 5 questionnaires showed good results) affecting to internal consistency results. The main issue we face concerns the lack of evidence regarding the unidimensionality of the scale, which does not seem to hold due to the presence of four different subdimensions. Furthermore, there is insufficient information to determine whether these subdimensions are truly distinct. This similarly occurs in the Connor–Davidson Resilience Scale (CD-RISC), where high levels of apparent internal consistency also coexist with an unstable factorial structure [78]. In short, this situation shows that the questionnaire needs substantial changes—both in the constructs it is meant to measure and in the items themselves—because its structural validity is not adequate and requires greater precision. These changes should lead to a revised version of the instrument, as was performed with BARQ-R.

### 3.5. MAIA, MAIA-2, and Brief MAIA-2

The iterative development from the Multidimensional Assessment of Interoceptive Awareness (MAIA) through the revised form (MAIA-2) to the Brief MAIA-2 presents a compelling narrative of the challenges inherent in measuring complex psychological constructs. The persistent instability across all versions, particularly in the subscales “not worrying” and “not distracting,” suggests a fundamental issue: these facets may not represent pure interoceptive awareness but rather a complex amalgam of awareness and emotional regulation strategies that are intrinsically intertwined and resist clean psychometric disentanglement. The improvements observed in structural validity, internal consistency, and convergent validity in MAIA-2 suggest that refining items can enhance statistical coherence. However, the persistence of structural inconsistencies indicates that the underlying theoretical model—proposing eight distinct dimensions—may oversimplify a more complex and multidimensional lived experience. Another possible reason for these poorer results may be the large number of items in these questionnaires, which can lead to a loss of precision and statistical validity [79]. The development of a brief version (Brief MAIA-2) prior to the full stabilization of the long-form structure risks cementing these conceptual ambiguities into a tool that is convenient yet potentially invalid. This trajectory mirrors the historical development of other multidimensional instruments like the Big Five Inventory, where the initial model of five independent personality factors was robustly supported by data, yet the specific facets within each broader domain (e.g., the components of “agreeableness”) often showed cross-loadings and cultural variability, requiring ongoing refinement and yielding brief versions that sometimes sacrifice nuanced validity for pragmatic utility [80]. Overall, MAIA, MAIA-2, and Brief MAIA-2 illustrate the persistent challenges of measuring a complex construct such as BA. Although MAIA-2 shows improvements in several psychometric properties, several subscales continue to exhibit instability, suggesting unresolved conceptual and psychometric issues.

### 3.6. PABE and PABE-QAG

Similar to what occurs with MAIA-2 and Brief MAIA-2, a fundamental hierarchy of psychometric evidence must be respected, wherein structural validity is a foundational property that must be robustly established prior to the investigation of other properties or other versions. Without a verified factorial structure that confirms that the instrument measures the intended theoretical domains, all subsequent short versions rest upon an uncertain and potentially flawed foundation [81]. The most pressing need is to resolve the fundamental questions concerning the instrument’s structural validity. More large-scale validation studies of the original PABE are needed to assess the real validity of the scale.

### 3.7. SBC

The SBC questionnaire was designed to measure body awareness and body dissociation. Psychometric evidence indicates that body awareness and bodily dissociation are separable—not strictly opposite—dimensions, and collapsing them into a single score obscures construct-specific variance. The inconsistent results were supported by the disparity in structural validity across the multiple adaptations, leaving it unclear whether the low consistency of the subscales stems from this issue or from another cause. Another possible reason could be that the cross-cultural adaptations made into languages other than the original were not carried out correctly, since in many of them we do not have that information. This issue has been observed in SF-36, where translation alone—without rigorous cross-cultural adaptation and invariance testing—failed to deliver conceptual equivalence [82]. Despite this, the convergent validity and responsiveness were very similar comparing subscales. It is necessary to obtain more precision and details in the factor analysis. In brief, SBC aims to assess both body awareness and bodily dissociation, but the inconsistent structural validity observed across adaptations raises concerns about whether these constructs can be meaningfully assessed within the same PROM.

### 3.8. Overall Discussion About All of the PROMs

In general, the reviewed PROMs for body awareness exhibit considerable heterogeneity in their psychometric properties. MAIA-2 demonstrates the most robust evidence, with improvements in structural validity, internal consistency, and convergent validity, although some subscales remain unstable and conceptual ambiguities persist. SBC may have similar and better results than MAIA-2 in the convergent validity of the two subscales, but these results should be used cautiously due to limited psychometric evidence and limited information being available regarding the translation and cross-cultural adaptation process in each country, as two studies carried out many of the adaptations. More methodologically rigorous studies with SBC could help to increase the level of evidence. On the other hand, promising results have been found for BARQ-R, which shows strengthened structural validity and promising reliability, but evaluation is limited to only two studies and cross-cultural adaptations are lacking.

Importantly, to better understand the heterogeneity among PROMs for body awareness, it is also important to consider the specific sub-constructs each instrument assesses. Some scales, such as MAIA-2, primarily focus on interoception and awareness of internal bodily signals, while others, like BARQ-R and BAQ, emphasize proprioception and bodily movement perception. Instruments such as SBC additionally incorporate aspects of embodiment and body dissociation, capturing the subjective sense of being in one’s body. Recognizing these distinctions helps to explain differences in psychometric performance and highlights that no single PROM fully captures the multidimensional nature of body awareness.

Based on the systematic review, we provide the following practical recommendations: MAIA-2 is suitable for assessing multidimensional interoceptive BA, while BARQ-R may be useful but more adaptations and studies are needed. SBC and BAQ can offer insights into BA and body dissociation but should be used cautiously due to limited psychometric evidence. For researchers, it is essential to select PROMs that align with the specific content of each scale and the sub-constructs of interest. Furthermore, more robust validation studies are needed to ensure good structural validity and to confirm the unidimensionality of each scale or subscale. This, together with a much more thorough translation and cultural adaptation process, will enhance the quality, interpretability, and reproducibility of body awareness outcomes.

### 3.9. Limitations and Strengths

This study has several limitations that should be considered when interpreting the findings. First, the construct of body awareness has been defined in multiple ways across the literature, which introduces potential variability in the validity and quality of the PROMs identified [72]. Second, the heterogeneity of the included populations—including differences in age, condition, and cultural backgrounds—limits the generalizability of the results. Third, variability in translation and cross-cultural adaptation procedures across studies may have affected the conceptual equivalence of PROMs and the measured properties, further complicating comparisons. Fourth, the methodological inconsistencies, such as the lack of reporting on key reliability metrics in some cases (e.g., ICC) or the absence of interdisciplinary expert involvement, reduced the robustness of the evidence in the studies. Moreover, the potential exclusion of relevant studies due to language restrictions or database limitations introduces the possibility of selection bias. These factors limit the robustness and comparability of the instruments. It would be advisable to include comparisons with measures other than PROMs, in order to examine their relationship with more objective assessments [83].

This study also has several strengths that deserve to be highlighted. Very extensive work has been carried out in this field after analyzing many studies and presenting to other researchers and professionals that PROMs are used for BA. In addition, an exhaustive analysis has been carried out following the COSMIN methodology. As regards the cross-cultural translation of the questionnaires into languages other than the original, many differences were found between them. These differences have been commented on following recommended guidelines and show mixed results. This work also serves to summarize and concentrate the quality of the translations so that each professional has complete information on the questionnaire in the language that they want to use with their patients.

## 4. Conclusions

The present review article examined a variety of questionnaires used to measure the category of BA, identifying twelve distinct questionnaires and more than thirty of their translations. Their psychometric properties were evaluated according to the COSMIN guidelines and the GRADE approach for PROMs instruments. The results reflect that the quality of psychometric properties varies considerably among the questionnaires. The findings suggest that most instruments may not fully capture the multidimensional nature of body awareness, due to conceptual complexity and the diversity of related sub-constructs such as interoception, proprioception, and embodiment. MAIA-2 currently appears to be the most acceptable instrument based on its comparatively better results and the quality of the evidence for its structural validity, internal consistency, and convergent validity, making it the most suitable instrument for research and clinical use at present. Careful selection of PROMs aligned with the specific content of each scale and the subscale of interest is recommended, and rigorous validation and cross-cultural adaptation processes should likewise be prioritized. Future studies must establish a homogeneous and well-defined conceptual framework of body awareness, as well as more detailed and methodologically improved translation and validation studies. These points will be crucial for the development of more precise, sensitive, and reliable measurement tools.

## Figures and Tables

**Figure 1 healthcare-13-03270-f001:**
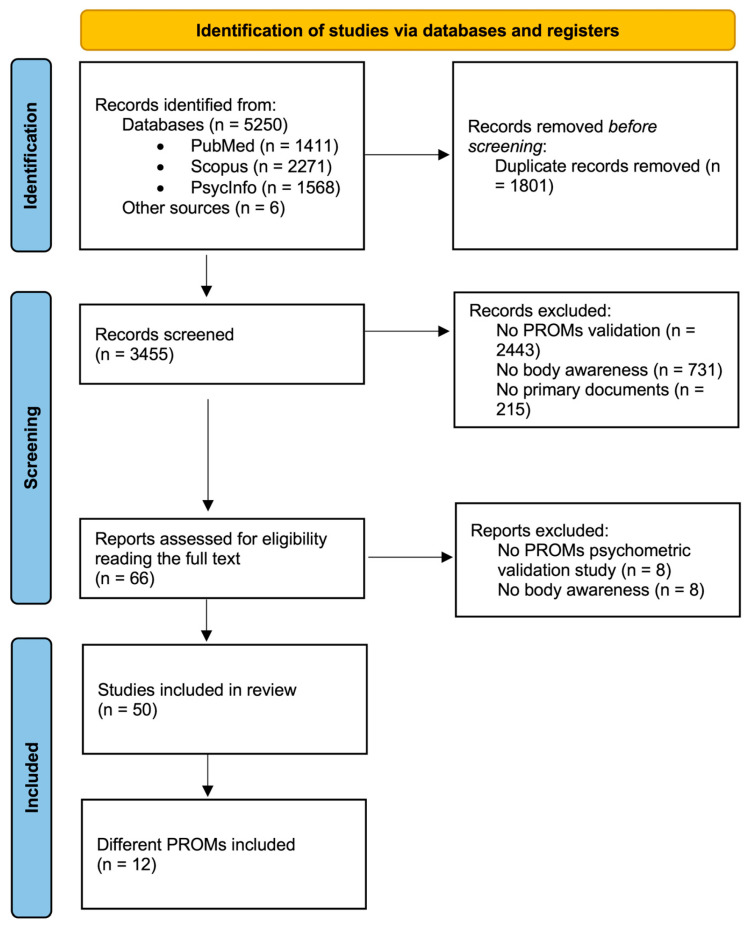
Flowchart of the selection process.

**Table 1 healthcare-13-03270-t001:** Description of the selected PROMs.

Name of the PROM	Original Language	Subscales	*n* of Items	Score Calculation	Translated Versions
Awareness-Body-Chart (ABC) [21]	German (Austria)	-	Body chart with 51 regions where you can choose a color based on perceived awareness	5-Likert scale. The higher the score, the lower the body awareness.	-
Body Awareness Rating Questionnaire (BARQ) [27]	Norwegian	Function, mood, feelings, and awareness	24 items	7-Likert scale. The higher the score, the lower the body awareness.	Turkish
Revised Body Awareness Rating Questionnaire (BARQ-R) [28]	Norwegian	-	12 items *	4-Likert scale. The higher the score, the lower the body awareness.	English (USA)
Body Awareness Questionnaire (BAQ) [29]	English (USA)	-	18 items	7-Likert scale. The higher the score, the lower the body awareness.	Sweden, German, Turkish, Spanish
Body Perception Questionnaire—Short Form (BPQ-SF) [12]	Spanish (Spain) and English (USA)	Body awareness, autonomic nervous, system reactivity	46 items	5-Likert scale. The higher the score, the less body awareness.	Italian, Chinese (China), Persian
Body Perception Questionnaire-Very short form (BPQ-VSF) [12]	Spanish (Spain) and English (USA)	-	12 items	Same as BPQ-VSF.	Chinese (China)
Multidimensional Assessment of Interoceptive Awareness (MAIA) [22]	English (USA)	Noticing, not distracting, not worrying, attention regulation, emotional awareness, self-regulation, body listening, trusting	32 items	6-Likert scale. The higher the score, the greater the body awareness.	Chinese (Taiwan), German (Germany), Greek, Japanese, Italian, Spanish (Chile), Spanish (Colombia), English (USA), Hungarian, Lithuanian, Malay, Portuguese
Multidimensional Assessment of Interoceptive Awareness Version 2 (MAIA-2) [26]	English (USA)	Same as MAIA	37 items	Same as MAIA.	French, Arabic (Lebanon), Chinese (China), Dutch (Netherlands), Norwegian, Persian
Brief Multidimensional Assessment of Interoceptive Awareness Version 2 (Brief MAIA-2) [25]	Polish	Same as MAIA	24 items	Same as MAIA.	-
Physical Activity Body Experiences Questionnaire (PABE) [23]	English (USA)	Mind/body connection, body acceptance	10 items	7-Likert scale. The higher the score, the lower the body awareness.	-
Physical Body Experiences Questionnaire Simplified for Active Aging (PBE-QAG) [30]	Italian	-	12 items	5-Likert scale. The higher the score, the lower the body awareness.	-
Scale of Body Connection (SBC) [24]	English (USA)	Body awareness, body dissociation	20 items	5-Likert scale. The higher the score, the lower the body awareness.	Italian, French, Dutch (Netherlands), Portuguese, English (USA), Hebrew (Israel), English (USA), French, English (Australia), Dutch (Netherlands), German, Italian, Spanish (Spain)

* A new short version of BARQ-R was conducted by Carpentier et al. (2024) with 9 items [31].

**Table 2 healthcare-13-03270-t002:** Study characteristics of PROMs validation studies.

PROM Name	Language	Psychometric Measures	Type of Participant	Mean Age	Female (%)	Type of Data Collection	Sample Size	Reliability Follow-Up	Intervention for Responsiveness
Awareness-Body-Chart (ABC)	Original: German (Austria) [21]	STRUV INTCO HTCOV CONV RELIA	Physiotherapy students with a good corporal consciousness state from the FH Joanneum University in Graz/Austria.	21 y (women) 22 y (men)	46%	Paper	172	Same day	-
Body Awareness Rating Questionnaire (BARQ)	Original: Norwegian [27]	STRUV INTCO CONV	Participants with long-lasting musculoskeletal and psychosomatic disorders and a small group of healthy people.	42.4 ± 11.1 y	77%	Paper	300	-	-
	Norwegian [32]	HTCOV RELIA RESPO MESER	Diagnosis of musculoskeletal pain and/or psychosomatic disorders with more than 3 months of duration.	42.2 ± 13.2 y	75.5%	Paper	50	2–7 days	6 months of Norwegian psychomotor physiotherapy
	Turkish [33]	INTCO RELIA HTCOV	Patients with CLBP who referred to the physiotherapy unit and healthy controls who were the patients’ caregivers.	44.9 ± 12.9 y (CLBP) 36.9 ± 11.9 y (healthy)	68.7% (CLBP) 50.5% (healthy)	?	99 (CLBP) 101 (healthy)	?	-
Revised Body Awareness Rating Questionnaire (BARQ-R)	Original: Norwegian [28]	STRUV RELIA INTCO MESER	People who had musculoskeletal and/or psychosomatic complaints who were referred to Norwegian psychomotor physiotherapy.	42.1 ± 11.0 y (Survey 2) 46.9 ± 12.0 y (Survey 3)	91% (Survey 2) 94% (Survey 3)	?	125 (Survey 2) 48 (Survey 3)	1 week	-
	English (USA) [31]	STRUV INTCO	Adults with and without self-reported musculoskeletal pain at the Minnesota State Fair and Highland Fest.	50.2 ± 17.2	66.1%	Paper and computerized	623	-	-
Body Awareness Questionnaire (BAQ)	Original: English (USA) [29]	STRUV INTCO RELIA HTCOV CONV	University students and no students with secondary school (high school) completed.	23.8 y	39% (students) 23% (non-students)	Paper	794	2 weeks	-
	Sweden [34]	STRUV INTCO	Patients with rheumatoid arthritis and nursing students with a good health status.	62 y	14% (patients) 6% (students)	Paper: letter	120 (patients) 120 (students)	-	-
	German (Germany) [35]	STRUV INTCO RELIA HTCOV RESPO	Patients with a condition of chronic pain (fibromyalgia, head pain, spinal pain, osteoarthritis, others).	50.3 ± 11.4 y	91.6%	?	512	10 weeks	Mindfulness, yoga, and qigong/Tai Chi
	Turkish [36]	STRUV INTCO RELIA HTCOV	Undergraduate students.	21.8 ± 2.3 y	45.0%	?	180	3 days	-
	Turkish [37]	INTCO RELIA HTCOV	Adults with non-specific low back pain and chronic of 3 months or more.	42.1 y	53.3%	?	180	7 days	-
	Spanish [38]	STRUV INTCO HTCOV	Adults.	25.3 y	64.7%	Computerized: online	281	-	-
	French [39]	STRUV INTCO HTCOV	University students and active workers.	30.5 ± 13.2 y	21.2%	Computerized: online	610	-	-
Body Perception Questionnaire—Short Form (BPQ-SF)	Original: Spanish (Spain) [12]	STRUV INTCO RELIA HTCOV CONV	Adults.	39.9 y	62%	Computerized: online	465	7 days	-
	Original: English (USA) [12]	STRUV INTCO	Healthy adults and undergraduate students.	35.1 y (USA residents) <20 y (USA students)	63% (USA residents) 47% (USA students)	Computerized: online (USA residents) Paper (USA students)	450 (USA residents) 315 (USA students)	-	-
	Italian [40]	STRUV INTCO	Adults.	34.7 y	59.2%	Computerized: online	493	-	-
	Chinese (China) [8]	STRUV INTCO RELIA HTCOV	Undergraduate students from the public university in China.	19.8 y	46.7%	Paper	688	4 weeks	-
	Persian [41]	STRUV INTCO HTCOV	Undergraduate students from the University of Tehran.	31.7 ± 51.3 y	57%	Computerized: online	748	-	-
Body Perception Questionnaire—Very Short Form (BPQ-VSF)	Original: Spanish (Spain) [12]	INTCO RELIA HTCOV CRITV	Same as BPQ-SF Spanish version.	-	-	-	-	-	-
	Original: English (USA) [12]	INTCO	Same as BPQ-SF English version.	-	-	-	-	-	-
	Chinese (China) [8]	INTCO RELIA HTCOV	Same as BPQ-SF Chinese version.					-	--
Multidimensional Assessment of Interoceptive Awareness (MAIA)	Original: English (USA) [22]	STRUV INTCO CONV	Students and instructors of body awareness components (yoga, tai chi…) or body-oriented psychotherapy.	42.2 y	86%	Computerized	325	-	-
	Chinese (Taiwan) [42]	STRUV INTCO RELIA HTCOV	Adults from social clubs that involved tai chi, qigong, martial arts, yoga, mindfulness, or meditation.	45.2 y	69.7%	Paper	294	2 weeks	-
	German (Germany) [43]	INTCO RELIA HTCOV RESPO	Adults.	42.6 ± 9.7 y	67.9%	Paper and computerized	1076	113 days	Daily practices of body scan and breath meditation
	Greek [44]	STRUV INTCO	Adults.	39.4 y	45.8%	Paper	107	-	-
	Japanese [45]	STRUV INTCO HTCOV	Adults and undergraduate students from Tokyo Kasei University and Sapporo international University.	20.2 y	67.7%	Paper	390	-	-
	Japanese [46]	STRUV INTCO RELIA HTCOV	Undergraduate students from Oita University, Osaka University, and Hiramatsu Oita College of Rehabilitation.	19.6 y	53%	Paper	268	2 weeks	-
	Italian [47]	STRUV INTCO HTCOV	Undergraduate students from the University G. d’Annunzio of Chieti.	20.5 y	42%	Paper	321	-	-
	Spanish (Chile) [48]	STRUV INTCO	Adults.	30.5 y	76.6%	Paper and computerized	470	-	-
	Spanish (Colombia) [49]	STRUV INTCO	Undergraduate university students from the Escuela Colombiana de Rehabilitación and the Universidad de Manizales.	21 y	64%	Paper	202	-	-
	English (USA) [50]	STRUV INTCO HTCOV	Eating disorder patients (adults and adolescents) from the University of California, Partial Hospital Program.	26.9 ± 9.9 y (adults) 15.1 ± 1.9 y (adolescents)	95.1% (adults) 93.8% (adolescents)	Computerized: electronic terminal	182 (adults) 194 (adolescents)	-	-
	Hungarian [51]	STRUV INTCO HTCOV	Adults.	35.7 ± 12.1 y	84.8%	Computerize: online	2109	-	-
	Lithuanian [52]	STRUV INTCO	Adults (most of them were undergraduate students).	21.9 ± 2.3 y	49%	?	386	-	-
	Malay [53]	STRUV INTCO	Adults.	33.8 y	49.4%	Paper	815	-	-
	Portuguese [54]	STRUV INTCO HTCOV RELIA	University students from the University of Évora.	23.3 y	86%	Paper	497	2 weeks	-
Multidimensional Assessment of Interoceptive Awareness Version 2 (MAIA-2)	Original: English (USA) [26]	STRUV INTCO CONV	Adult visitors of the Live Science residency project at the Science Museum of London.	30.6 y	47%	Paper	1090	-	-
	English (USA) [55]	STRUV INTCO RELIA	Southeastern United States college sample.	23.7 ± 7.2 y (Sample 1) 21.0 ± 1.3 y (Sample 2)	69.2% (Sample 1) 86.0% (Sample 2)	?	776	3 weeks	-
	French [56]	STRUV INTCO RELIA HTCOV	Adults recruited from websites and social media.	35 y	61%	Computerized: online	308	11 days	-
	Arabic (Lebanon) [57]	STRUV INTCO HTCOV	Adults.	22.7 y	59.9%	Computerized: online	359	-	-
	Chinese (China) [58]	STRUV INTCO HTCOV	Adults from Zhejiang University.	21.6 y	61.7%	?	627	-	-
	Dutch (Netherlands) [59]	STRUV INTCO RELIA	University students from the University Medical Center Groningen.	35.1 ± 15.7 y	68.6%	Computerized: online	1054	-	-
	Norwegian [60]	STRUV INTCO HTCOV	Adults from two Norwegian municipalities, two Norwegian Universities, and online recruitment.	41–45 y	81%	Computerized: online	306	-	-
	Persian [61]	STRUV INTCO HTCOV	Adults from Shahid Beheshti University’s social platforms and students’ social media groups.	31.8 ± 7.1 y	86.1%	Computerized: online	475	-	-
	Spanish (Peru) [62]	STRUV INTCO	University and military school adults.	?	61%	Computerized: online	414	-	-
Brief Multidimensional Assessment of Interoceptive Awareness Version 2 (Brief MAIA-2)	Polish [25]	STRUV INTCO	University students of Physical Education and elite athletes in speed skating.	26.1 ± 9.1 y	54.8%	Paper	323	-	-
Physical Activity Body Experiences Questionnaire (PABE)	Original: English (USA) [23]	STRUV INTCO RELIA HTCOV CONV	Undergraduates from the University of South Florida.	21.0 ± 5.2 y (Sample 1) 21.0 ± 5.0 y (Sample 2) 19.9 ± 2.2 y (Sample 3)	100%	Computerized: online	664	-	-
Physical Body Experiences Questionnaire Simplified for Active Aging (PBE-QAG)	Original: Italian [30]	STRUV INTCO	Elderly people at the University of the Third Age of Cagliari.	68.8 y	59.4%	?	106	-	-
	English (USA) [63]	STRUV	Adults and older adults (healthy or with stroke) through the Minnesota State Fair and a previous project.	70.3 ± 4.8 y (older adults without stroke) 50.0 ± 16.8 y (adults without stroke) 58.0 ± 12.9 y (adults with stroke)	57.8% (older adults without stroke) 63.7% (adults without stroke) 30.5% (adults with stroke)	Computerized: electronic terminal	133 (older adults without stroke) 530 (adults without stroke) 36 (adults with stroke)	-	-
Scale of Body Connection (SBC)	Original: English (USA) [24]	STRUV INTCO CONV	Undergraduate students.	20 y	55%	Paper	291	-	-
	Portuguese (Portugal) [64]	STRUV INTCO HTCOV	Adults.	34.08 ± 11.74 y	49%	Computerized: online	909	-	-
	Portuguese (Portugal) [65]	STRUV INTCO	Adults.	31.0 y	49.0%	Computerized: online	909	-	-
	Italian [65]	STRUV INTCO	Adults.	27.0 y	68.8%	Computerized: online	576	-	-
	French [65]	STRUV INTCO	Adults.	39.0 y	31.4%	Computerized: online	198	-	-
	Dutch (Netherlands) [65]	STRUV INTCO	Undergraduate students.	20.0 y	75.5%	Paper	434	-	-
	English (USA) [65]	STRUV INTCO	Undergraduate students, lesbian women, and somatic therapists.	20.0 y (students) 48.5 y (lesbian women) 49.0 y (therapists)	57.7% Paper (students) 100% (lesbian women) 78.7% (therapists)	Paper (students); computerized: online (therapists and lesbian women)	291 290 328	-	-
	Hebrew (Israel) [65]	STRUV INTCO	Undergraduate students.	28.0 y	62%	Paper	608	-	-
	English (USA) [66]	HTCOV RESPO	Trauma, overweight/obese, and substance-use patients.	?	100%	?	152 (trauma) 24 (overweight) 99 (substance use) 24 (trauma)	-	Yoga, mindfulness meditation, or body awareness
	French [66]	HTCOV RESPO	Eating disorder patients.	?	100%	?	12	-	Mindfulness meditation
	Dutch (Netherlands) [66]	RESPO	Chronic pain and mood disorder patients.	?	82% (chronic pain) 78% (mood disorders)	?	50 (chronic pain) 164 (mood disorders)	-	Body awareness
	German [66]	RESPO	Colorectal cancer, chronic pain, major depression, and neck pain patients.	?	79% (cancer) 91–93% (chronic pain) 76% (depression) 81% (neck pain)	?	27 (cancer) 512 (chronic pain) 21 (depression) 22 (neck pain)	-	Yoga, body awareness, or bodywork
	Italian [67]	STRUV INTCO HTCOV	Adults.	30.4 ± 9.4 y	68.7%	Computerized: online	576	-	-
	Spanish (Spain) [68]	STRUV INTCO RELIA HTCOV	Adults from mindfulness and meditation associations.	41.3 ± 11.2 y	61.9%	Computerized: online	578	1 month	-

CONV, content validity; STRUV, structural validity; INTCO, internal consistency; RELIA, reliability; MESER, measurement error; CRITV, criterion validity; HTCOV, hypothesis testing for construct validity; RESPO, responsiveness; CLBP, chronic low back pain; ?, Information not reported.

**Table 3 healthcare-13-03270-t003:** Pooled psychometric results, and quality of the evidence analysis.

		Structural Validity	Internal Consistency	Reliability	Hypotheses Testing for Construct Validity			Criterion Validity	Responsiveness	Measurement Error
					Convergent	Divergent	Known			
Body Awareness Rating Questionnaire (BARQ)	n of outcomes	1	3	2	2	2				1
	Summary of the risk of bias	100% Doubtful	100% Doubtful	50% Adequate, 50% Inadequate	100% Adequate	100% Adequate				100% Adequate
	Pooled result	−(Total variance explained = 46.9%)	+(α = 0.65–0.96)	+(ICC = 0.74–0.96)	−(r = 0.02, 0.53)	±(r = −0.39, 0.26)				? (SDC = 0.69–0.85)
	Quality of evidence	n.a.	Low	Very low	Moderate	Moderate				n.a.
Revised Body Awareness Rating Questionnaire (BARQ-R)	n of outcomes	2	2	1						1
	Summary of the risk of bias	100% Very good	100% Very good	100% Very good						100% Adequate
	Pooled result	+(X^2^ = 13.6–205.7; Fit residual = −2.01 to 0.7)	+(PSI = 0.72–0.76)	+(ICC = 0.83)						? (SDC = 6.26)
	Quality of evidence	High	High	n.a.						n.a.
Body Awareness Questionnaire (BAQ)	n of outcomes	6	7	4	6	2	2		1	
	Summary of the risk of bias	33% Very good, 33% Adequate, 33% Doubtful	43% Very good, 57% Doubtful	75% Very good, 25% Doubtful	50% Very good, 50% Adequate	50% Very good, 50% Adequate	50% Adequate, 50% Doubtful		100% Adequate	
	Pooled result	−(CFI = 0.68–0.95; RMSEA = 0.04–0.09)	+(α = 0.72–0.91)	+(ICC = 0.80–0.86)	−(r = 0.00, 0.80)	±(r = −0.01, −0.75)	+(*p* = 0.001–0.03)		+(*p* < 0.001)	
	Quality of evidence	Low	Low	Low	Low	Moderate	Low		n.a.	
Body Perception Questionnaire—Short Form (BPQ-SF)	n of outcomes	5	5	2	3	1				
	Summary of the risk of bias	60% Very good, 40% Doubtful	100% Very good	50% Very good, 50% Doubtful	100% Adequate	100% Adequate				
	Pooled result	±(CFI = 0.90–0.98; RMSEA = 0.02–0.11)	+(α/ω = 0.77–0.96)	+(ICC = 0.71–0.99)	±(r = 0.24, 0.69)	−(r = 0.11, 0.19)				
	Quality of evidence	Moderate	High	Low	High	n.a.				
Body Perception Questionnaire—Very Short Form (BPQ-VSF)	n of outcomes		3	2	2	1		1		
	Summary of the risk of bias		100% Very good	50% Very good, 50% Doubtful	100% Adequate	100% Adequate		100% Adequate		
	Pooled result		+(α = 0.83–0.91)	±(ICC/r = 0.68–0.97)	±(r = 0.28, 0.55)	−(r = 0.18)		+(r = 0.94)		
	Quality of evidence		High	Low	High	n.a.		n.a.		
Multidimensional Assessment of Interoceptive Awareness (MAIA)	n of outcomes	13	14	4	6	7	2		1	
	Summary of the risk of bias	47% Very good, 8% Adequate, 30% Doubtful, 15% Inadequate	93% Very good, 7% Inadequate	75% Doubtful, 25% Inadequate	100% Very good	72% Very good, 14% Adequate, 14% Doubtful	100% Adequate		100% Very good	
	Pooled result	±(CFI = 0.82–0.98; RMSEA = 0.02–0.07; SRMR = 0.04–0.09)	±(α/ω = 0.40–0.96)	±(ICC = 0.52–0.85)	−(r = −0.49, 0.64)	−(r = −0.66, 0.60)	±(*p* = 0.001–0.56)		(±) (*p* = 0.001–0.44)	
	Quality of evidence	Moderate	High	Very low	Moderate	Moderate	High		n.a.	
Multidimensional Assessment of Interoceptive Awareness Version 2 (MAIA-2)	n of outcomes	9	9	3	5	2				
	Summary of the risk of bias	66% Very good, 34% Doubtful	100% Very good	34% Very good, 66% Doubtful	60% Very good, 20% Adequate, 20% Doubtful	100% Very good				
	Pooled result	±(CFI = 0.86–0.95; RMSEA = 0.05–0.11; SRMR = 0.05–0.10)	+(α/ω = 0.58–0.93)	±(ICC = 0.63–0.82)	±(r = −0.07, 0.64)	−(r= −0.50, 0.27)				
	Quality of evidence	Moderate	High	Low	Moderate	High				
Physical Body Experiences Questionnaire Simplified for Active Aging (PBE-QAG)	n of outcomes	2	1							
	Summary of the risk of bias	50% Very good, 50% Doubtful	100% Very good							
	Pooled result	−(CFI =0.989; RMSEA = 0.076)	+(α = 0.65–0.89)							
	Quality of evidence	Low	n.a.							
Scale of Body Connection (SBC)	n of outcomes	10	9	1	5	1			4	
	Summary of the risk of bias	10% Very good, 90% Doubtful	89% Very good, 11% Doubtful	100% Doubtful	40% Very good, 40% Adequate, 20% Doubtful	100% Very good			100% Very good	
	Pooled result	±(CFI = 0.74–0.99; RMSEA = 0.02–0.10; SRMR = 0.07–0.08)	+(α = 0.62–0.86)	±(r = 0.67, 0.76)	±(r = 0.02–0.65)	+(r = 0.45, 0,64)			±(*p* = 0.001–0.66)	
	Quality of evidence	Very low	Moderate	n.a.	Low	n.a.			Moderate	

+, sufficient; −, insufficient; ±, inconsistent; ?, indeterminate; n.a., not applicable.

## Data Availability

No new data were created or analyzed in this study.

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
