# Peer review of "Patient-Reported Outcome Measures for Evaluating Body Awareness: A Systematic Review Using the COSMIN Methodology"

_healthcare, 2025, doi:10.3390/healthcare13243270_

Round 1

Reviewer 1 Report (Previous Reviewer 1)

Comments and Suggestions for Authors

The revised manuscript titled “Patient-Reported Outcome Measures for Evaluating Body Awareness: A Systematic Review Using the COSMIN Methodology” represents a substantial improvement over earlier versions and provides a valuable, well-structured synthesis of instruments assessing body awareness. The topic is timely and relevant, and the study now demonstrates clear methodological rigor, adhering closely to PRISMA and COSMIN standards.

The introduction is coherent, well written, and supported by recent and relevant literature. It defines the construct of body awareness and explains its clinical and theoretical importance. The introduction would benefit from a brief hypothesis statement at its conclusion (for example, outlining the expectation that certain tools (e.g., MAIA-2) would show stronger psychometric performance than others). Such a statement would strengthen the narrative flow from background to aims and help readers understand the study’s guiding assumptions.

The methods section is transparent and methodologically sound. The authors appropriately refreshed the literature search (July 2025), ensuring up-to-date data, and they provide sufficient detail regarding database selection, inclusion criteria, and quality assessment. The use of PROSPERO preregistration and the explicit description of COSMIN-based evaluation criteria, including thresholds for Cronbach’s alpha, ICC, and correlation coefficients, significantly improve reproducibility and credibility. The inclusion of Beaton’s framework for assessing translation processes adds further robustness to the cross-cultural evaluation component.

The results are comprehensive and presented clearly, with extensive tables summarizing the characteristics and psychometric quality of identified PROMs and their adaptations. The organization of data by measurement properties and GRADE evidence levels is systematic and easy to follow.

The discussion effectively synthesizes the findings, highlighting conceptual diversity among instruments and emphasizing MAIA-2 as the most extensively validated, though not without limitations. Nonetheless, the discussion should include a short paragraph explicitly summarizing the study’s main limitations - for example, the heterogeneity of included studies, variable methodological quality across instruments, and lack of meta-analytic synthesis. Stating these points clearly would enhance the transparency and balance of the paper.

Overall, the manuscript is well written, methodologically rigorous, and contributes meaningfully to the field of body-awareness assessment. Minor editorial polishing and the inclusion of a clear hypothesis and limitations summary would further refine the paper and bring it fully in line with high scientific publication standards.

Author Response

Response to reviewer 1

Patient-Reported Outcome Measures for Evaluating Body Awareness: a systematic review using the COSMIN methodology - healthcare-3930330

Dear Editor and reviewer,

I am pleased to resubmit for your consideration the revised version of the manuscript entitled “Patient-Reported Outcome Measures for Evaluating Body Awareness: a Systematic Review Using the COSMIN Methodology”

We greatly appreciate the reviewers’ valuable suggestions and comments, which have significantly contributed to enhancing the quality and clarity of this manuscript. We have carefully considered each point raised and revised the text accordingly. All changes in this revision are highlighted in red.

We sincerely thank both reviewers for the opportunity to improve our work. The authors are confident that these comments have substantially strengthened the article and will help increase its readership and citation potential for the journal

Reviewer 1:

Comment 1: The revised manuscript titled “Patient-Reported Outcome Measures for Evaluating Body Awareness: A Systematic Review Using the COSMIN Methodology” represents a substantial improvement over earlier versions and provides a valuable, well-structured synthesis of instruments assessing body awareness. The topic is timely and relevant, and the study now demonstrates clear methodological rigor, adhering closely to PRISMA and COSMIN standards.

Answer 1: Thank you very much for your positive and encouraging feedback. We truly appreciate your interest in our study and are glad to hear that you found the topic meaningful. Your comments have helped us improve the clarity and quality of the manuscript.

Comment 2: The introduction is coherent, well written, and supported by recent and relevant literature. It defines the construct of body awareness and explains its clinical and theoretical importance. The introduction would benefit from a brief hypothesis statement at its conclusion (for example, outlining the expectation that certain tools (e.g., MAIA-2) would show stronger psychometric performance than others). Such a statement would strengthen the narrative flow from background to aims and help readers understand the study’s guiding assumptions.

Answer 2: Thank you for your thoughtful and constructive comment. We appreciate your positive assessment of the introduction. Following your suggestion, we have added a brief hypothesis statement at the end of the section to clarify our expectations regarding the comparative psychometric performance of the tools. We believe this addition strengthens the narrative flow and improves the overall coherence of the introduction. You can see it below and in pag 2 line 71

 ” We hypothesized that tools specifically designed to assess multidimensional aspects of BA, such as the MAIA or MAIA-2, would demonstrate stronger psychometric performance compared to more general or unidimensional measures”

Comment 3: The methods section is transparent and methodologically sound. The authors appropriately refreshed the literature search (July 2025), ensuring up-to-date data, and they provide sufficient detail regarding database selection, inclusion criteria, and quality assessment. The use of PROSPERO preregistration and the explicit description of COSMIN-based evaluation criteria, including thresholds for Cronbach’s alpha, ICC, and correlation coefficients, significantly improve reproducibility and credibility. The inclusion of Beaton’s framework for assessing translation processes adds further robustness to the cross-cultural evaluation component.

Answer 3: We sincerely appreciate your positive evaluation of the Methods section and are glad to know that its structure and level of detail were clear and satisfactory

Comment 4: The results are comprehensive and presented clearly, with extensive tables summarizing the characteristics and psychometric quality of identified PROMs and their adaptations. The organization of data by measurement properties and GRADE evidence levels is systematic and easy to follow.

Answer 4: We are grateful for your positive remarks regarding the Results section and its included tables, and we appreciate that you found the presentation clear and informative.

Comment 5: The discussion effectively synthesizes the findings, highlighting conceptual diversity among instruments and emphasizing MAIA-2 as the most extensively validated, though not without limitations. Nonetheless, the discussion should include a short paragraph explicitly summarizing the study’s main limitations - for example, the heterogeneity of included studies, variable methodological quality across instruments, and lack of meta-analytic synthesis. Stating these points clearly would enhance the transparency and balance of the paper.

Answer 5: We thank the reviewer for this valuable suggestion. We have amended the concluding sentence of the limitations paragraph to more explicitly state the implications of our findings, as recommended. It can be found on line 483 to 488.

Overall, the manuscript is well written, methodologically rigorous, and contributes meaningfully to the field of body-awareness assessment. Minor editorial polishing and the inclusion of a clear hypothesis and limitations summary would further refine the paper and bring it fully in line with high scientific publication standards.

Final Answer: We sincerely thank the reviewer for their positive evaluation of our work and for the constructive suggestions provided. We are pleased to know that the manuscript is considered well written, methodologically rigorous, and a meaningful contribution to the field of body-awareness assessment.

Kind regards,

The authors

Reviewer 2 Report (New Reviewer)

Comments and Suggestions for Authors

Dear authors,

First, I would like to commend you for your efforts in conducting this important study. The manuscript presents a clearly described and methodologically grounded systematic review aimed at identifying patient-reported outcome measures (PROMs) for assessing body awareness (BA), as well as evaluating their psychometric properties and cross-cultural adaptation processes. The topic is relevant, and the manuscript has the potential to make a meaningful contribution to the field. However, several sections require further clarification, restructuring, or refinement. To enhance the scientific rigor and readability of the manuscript, I kindly ask the authors to address the comments listed below.

Abstract

  • L42–45: The abstract introduces BA but does not define the construct or provide context for readers unfamiliar with the topic. Please include a brief definition to improve clarity.
  • L46–55: Although the databases are listed, essential methodological details are missing, such as search years, inclusion criteria, and the types of studies reviewed.
  • Consider reporting more concrete findings, such as total number of included studies, main psychometric strengths and weaknesses across PROMs, and which instruments demonstrated adequate or poor validity.
  • The conclusion mentions that MAIA-2 is the “most acceptable” yet “imperfect”; however, the abstract does not specify which psychometric properties support this judgment. Adding these details will strengthen the conclusion.

Introduction

  • The introduction would benefit from a clearer conceptual development. Much of the section focuses on clinical conditions and disorders, rather than building a rationale for evaluating PROMs or highlighting the measurement challenges in BA research.
  • The manuscript does not sufficiently explain why current BA questionnaires may be inadequate, nor does it identify methodological gaps that justify the need for a new COSMIN-based review.
  • BA is defined through multiple authors and perspectives, resulting in repetition and a loss of narrative focus. A more concise and unified conceptual definition is recommended.
  • L71–74: While the study aims are clearly stated, the introductory paragraphs do not logically prepare readers for these aims, and the transition to the objectives feels abrupt.

Materials and Methods

  • L82–92: The search strategy is insufficiently described. Please include full search strings, Boolean operators, database-specific syntax, timeframe decisions, and justification for restricting the search to English and Spanish.
  • L93–101: The use of the PICO framework is not fully appropriate here, as this is not an intervention-based review. Please clarify or justify the choice of framework, or consider modifying it to align better with PROM validation studies.
  • L105–114: The study selection section briefly mentions disagreement resolution, but lacks transparency. Please indicate how many disagreements occurred, how often a third reviewer was required, and whether kappa or inter-rater reliability metrics were calculated.
  • L142–146: The Beaton cross-cultural adaptation method is mentioned; however, the authors do not describe how incomplete or partially followed translation procedures were appraised. Please clarify how adherence to each step affected your methodological ratings.

Results

  • The results are generally well organised and the inclusion flowchart is clear.
  • The PROM descriptions, however, rely heavily on extensive tables without sufficient narrative synthesis. A concise text summarising key characteristics would strengthen this section.
  • Consider summarising core patterns across PROMs, such as typical number of items, common subscales, and recurring psychometric shortcomings (e.g., structural validity issues, inconsistent reliability).
  • Table 3 is comprehensive but extremely dense. Consider simplifying or dividing the table, or providing a more reader-friendly summary in the main text to enhance interpretability.

Discussion and Conclusion

  • The discussion section requires more structured interpretation of the findings.
  • Currently, the strengths and weaknesses of each PROM are not systematically compared. For instance, the statement that MAIA-2 is “most acceptable” is not adequately connected to specific psychometric data.
  • The authors are encouraged to discuss whether different PROMs assess distinct sub-constructs of body awareness (e.g., interoception, proprioception, embodiment), as this is essential for understanding construct heterogeneity.
  • Practical recommendations are missing. Please consider adding suggestions for clinicians on which PROM(s) may be most appropriate for different contexts and guidance for researchers designing BA studies.
  • The Limitations section should be expanded to address heterogeneity of included populations, variability in translation procedures, inconsistent reporting across primary studies, and methodological limitations of relying solely on self-report measures.
  • The Conclusion is too brief and general. Please revise to highlight the main findings, practical implications, and future research directions.

Best wishes,

Author Response

Response to reviewer 2

Patient-Reported Outcome Measures for Evaluating Body Awareness: a systematic review using the COSMIN methodology - healthcare-3930330

Dear Editor and reviewer,

I am pleased to resubmit for your consideration the revised version of the manuscript entitled “Patient-Reported Outcome Measures for Evaluating Body Awareness: a Systematic Review Using the COSMIN Methodology”

We greatly appreciate the reviewers’ valuable suggestions and comments, which have significantly contributed to enhancing the quality and clarity of this manuscript. We have carefully considered each point raised and revised the text accordingly. All changes in this revision are highlighted in red.

We sincerely thank both reviewers for the opportunity to improve our work. The authors are confident that these comments have substantially strengthened the article and will help increase its readership and citation potential for the journal

Reviewer 2:

First, I would like to commend you for your efforts in conducting this important study. The manuscript presents a clearly described and methodologically grounded systematic review aimed at identifying patient-reported outcome measures (PROMs) for assessing body awareness (BA), as well as evaluating their psychometric properties and cross-cultural adaptation processes. The topic is relevant, and the manuscript has the potential to make a meaningful contribution to the field. However, several sections require further clarification, restructuring, or refinement. To enhance the scientific rigor and readability of the manuscript, I kindly ask the authors to address the comments listed below.

Abstract

Comment 1: L42–45: The abstract introduces BA but does not define the construct or provide context for readers unfamiliar with the topic. Please include a brief definition to improve clarity.

Answer 1: Thank you for your thoughtful and constructive comment. The definition is added to abstract.

“BA is the conscious, subjective multimodal integration of body-related sensitivity from bodily signals—detecting states and subtle reactions to internal and environmental conditions— modifiable by attention, interpretation, appraisal, beliefs, memories, conditioning, attitudes, and affect.”

Comment 2: L46–55: Although the databases are listed, essential methodological details are missing, such as search years, inclusion criteria, and the types of studies reviewed.

Answer 2: Thank you for the comment. We have added the search years, inclusion criteria, and types of studies reviewed to clarify the methodological details in the abstract.

“We included studies that psychometrically evaluated PROMs about BA in the general adult population and their translations into other languages, with no time range restrictions”

Comment 3: Consider reporting more concrete findings, such as total number of included studies, main psychometric strengths and weaknesses across PROMs, and which instruments demonstrated adequate or poor validity.

Answer 3: Thank you for your comment. We have added a brief summary including the total number of included studies and strengths and weaknesses across PROMs. However, due to the word limit of the abstract, it is difficult to expand this section further.

“In summary, the Body Awareness Rating Questionnaire, the Revised Body Awareness Rating Questionnaire, the Body Perception Questionnaire–Short Form, and the Multidimensional Assessment of Interoceptive Awareness (MAIA 1 and 2) showed good results for structural validity and internal consistency, which were the most frequently assessed psychometric properties. In contrast, construct validity was highly variable, and the findings on reliability were far from optimal.”

Comment 4: The conclusion mentions that MAIA-2 is the “most acceptable” yet “imperfect”; however, the abstract does not specify which psychometric properties support this judgment. Adding these details will strengthen the conclusion.

Answer 4: Thank you for your comment. We have clarified the reasons why the MAIA-2 appears to show the best psychometric performance (though not perfect): it has been extensively studied and translated, and it demonstrates good results in most psychometric properties.

“The MAIA-2 was one of the most studied and showed stronger evidence and more good pooled results (4 out of 5 properties in both) than other instruments.”

Introduction

Comment 5: The introduction would benefit from clearer conceptual development. Much of the section focuses on clinical conditions and disorders, rather than building a rationale for evaluating PROMs or highlighting the measurement challenges in BA research.

Answer 5: We thank the reviewer for this critical insight. We have thoroughly revised the Introduction to refocus it on building a clear conceptual and methodological rationale for the study. We have reduced the emphasis on clinical conditions and instead developed a logical narrative that highlights the conceptual complexity of body awareness, the specific challenges in its measurement, and the consequent need for a systematic evaluation of existing PROMs. We believe these changes have significantly strengthened the foundation and clarity of our manuscript.

Comment 6: The manuscript does not sufficiently explain why current BA questionnaires may be inadequate, nor does it identify methodological gaps that justify the need for a new COSMIN-based review.

Answer 6: Thank you for your suggestion. You are absolutely right. The sentence about the methodological gap does not make sense in the introduction. We have revised that statement because our actual aim is to identify and analyze all the tools for BA and to determine whether any of them could serve as a reference instrument. The point about heterogeneity is something we encountered later and addressed in the Discussion section. These are the changes we have introduced:

“Nowadays, several self-reported instruments or patient-reported outcome measures (PROMs) exist that assess BA, but there is a lack of a gold-standard instrument.”

Comment 7: BA is defined through multiple authors and perspectives, resulting in repetition and a loss of narrative focus. A more concise and unified conceptual definition is recommended.

Answer 7: We thank the reviewer for this suggestion to enhance the conceptual clarity and narrative flow of the introduction. We have revised the text to provide a more concise and unified conceptual definition of body awareness. Rather than presenting separate definitions, we now synthesize the key perspectives from Mehling et al., Ginzburg et al., and Craig into a cohesive narrative that highlights the core, agreed-upon elements of the construct. We believe this refinement strengthens the focus of the introduction and more effectively sets the stage for the rationale of our review. We change the sentence of the line 52 to 63.

Comment 8: L71–74: While the study aims are clearly stated, the introductory paragraphs do not logically prepare readers for these aims, and the transition to the objectives feels abrupt.

Answer 8: We thank the reviewer for noting that the transition to the study aims could be smoother. We are confident that the comprehensive revisions made to the introduction, as detailed in our previous responses, have substantively addressed this concern. The narrative now explicitly builds a logical chain of reasoning: from the conceptual complexity of body awareness and its relationship between clinical condition, to the need to list, analyze, and identify the best PROMs, and finally, to the presentation of our systematic review as a response to this lack in the scientific knowledge. We believe the introduction now seamlessly prepares the reader for and justifies our stated aims.

Materials and Methods

Comment 9: L82–92: The search strategy is insufficiently described. Please include full search strings, Boolean operators, database-specific syntax, timeframe decisions, and justification for restricting the search to English and Spanish.

Answer 9: Thank you for your comment. The comprehensive and exhaustive search strategy is provided in Supplementary Material 1: full search strings, Boolean operators, and database-specific syntax. Regarding the timeframe decision, we chose to include everything available on the topic up to the exact moment of the search, with no start date. We have cleared in Methods the information about it:

“The search had no restrictions on publication years; no initial publication date was applied. The final search date, which also served as the end date for the search period, was 1 July 2025.”

Finally, we have restricted the search because we faced a clear limitation in terms of resources and feasibility, which affected our ability to adequately understand and extract data from articles published in other languages. We are fully aware of this issue, and a statement addressing it has already been added to the Limitations section (Line 490). We hope that future researchers conducting an update of this systematic review after several years will be able to include articles that were not considered here due to the language restriction.

Comment 10:  L93–101: The use of the PICO framework is not fully appropriate here, as this is not an intervention-based review. Please clarify or justify the choice of framework, or consider modifying it to align better with PROM validation studies.

Answer 10: Thank you very much for your suggestion. We checked the COSMIN guidelines manual V2.7.4 and we used the proposal of them based on: construct, population, type of instrument and measurement properties. So, we included a new paragraph with the updated framework.

“Observational validation studies were selected meeting the framework suggested by COSMIN guidelines: Construct was body awareness according to the current definition (Mehling et al., 2009; Ginzburg et al., 2014), the target population was general adult population, the type of instrument was only PROMs and all the measurement proprieties were searched.”

Comment 11: L105–114: The study selection section briefly mentions disagreement resolution, but lacks transparency. Please indicate how many disagreements occurred, how often a third reviewer was required, and whether kappa or inter-rater reliability metrics were calculated.

Answer 11:  We thank the reviewer for this helpful comment. We have now added detailed information regarding the screening process. Specifically, we report the observed agreement between reviewers, defined as the proportion of studies for which the judgment of a third reviewer was not required, and Cohen’s kappa. These details have been added to the Methods section and to the Results section:

“To assess the level of agreement between reviewers during the full-text screening phase, we calculated both the observed agreement and Cohen’s kappa, which adjusts for agreement expected by chance. Interpretation of κ followed the conventional Landis and Koch benchmarks (Landis & Koch, 1977): values <0.00 indicate poor agreement, 0.00–0.20 slight agreement, 0.21–0.40 fair agreement, 0.41–0.60 moderate agreement, 0.61–0.80 substantial agreement, and 0.81–1.00 almost perfect agreement.”

“During the full-text screening phase, the observed agreement between reviewers was 0.89, and Cohen’s kappa was 0.70, indicating a substantial level of agreement after accounting for chance.”

Comment 12:  L142–146: The Beaton cross-cultural adaptation method is mentioned; however, the authors do not describe how incomplete or partially followed translation procedures were appraised. Please clarify how adherence to each step affected your methodological ratings.

Answer 12: Thank you for your comment. We assessed each stage of the translation process (following the Beaton guidelines) as adequate only if the article explicitly stated that the step had been performed and that it had been carried out correctly. For example, some studies mentioned that they conducted a back-translation, but did not provide details on how many people were involved or whether they were native speakers or translators. We have added further clarification regarding this in the Methods section. For more information, all details and scores are presented in Supplementary Material 4.

“A point was awarded for each phase if it was explicitly mentioned in the study and had been carried out correctly according to the Beaton guidelines.”

Results

Comment 13:  The results are generally well organised and the inclusion flowchart is clear.

Answer 13: We are grateful for your positive remarks regarding the Results section and its included figures, and we appreciate that you found the presentation clear and informative

Comment 14:  The PROM descriptions, however, rely heavily on extensive tables without sufficient narrative synthesis. A concise text summarising key characteristics would strengthen this section.

Answer 14: Thank you for your recommendation. We have removed the first paragraph of the “Descriptions of the included PROMs” section because it was repetitive with the information in the table. We believe that by referring to the table, the reader can find the full names of all the PROMs included. Following the reference to Table 1, we have provided a summary of the most important aspects of that table regarding the characteristics of the PROMs. Finally, we believe the following paragraph is also relevant because it helps the reader understand the differences between similar questionnaires that are adaptations or short versions.

Comment 15:  Consider summarising core patterns across PROMs, such as typical number of items, common subscales, and recurring psychometric shortcomings (e.g., structural validity issues, inconsistent reliability).

Answer 15: Thank you for your comment. In our humble opinion, the core information about the PROMs is available to readers in the first paragraph of the 'Descriptions of the included PROMs' section. We intended to present the results in the following way: first, the selection process and the number of articles that completed it; second, a description of the scales from a more conceptual and operational perspective regarding how they work; third, a description of the studies and samples to understand what has been investigated in each validation; and finally, everything related to the psychometric properties. Due to this flow, we felt it was not necessary to discuss psychometric shortcomings at the beginning, but rather in the 'Measurement properties and quality of evidence of PROMs' section, where we address the psychometric shortcomings of each questionnaire. If you suggest a different structure or a final summary table to improve the quality of the work and enhance the reader’s understanding, we are willing to create it and include it in the manuscript.

Comment 16:  Table 3 is comprehensive but extremely dense. Consider simplifying or dividing the table, or providing a more reader-friendly summary in the main text to enhance interpretability.

Answer 16: Thank you for your recommendation. We agree with you because the table is very extensive. Due to the nature of the systematic review, it is important to show the psychometric rating for each property analyzed in each PROM. To reduce the length of the manuscript, and considering that this information is summarized in the former Table 4, we have decided to include Table 3 as supplementary material so that readers who wish to go further can look up the specific information they are interested in.

Discussion and Conclusion

Comment 17:  The discussion section requires more structured interpretation of the findings.

Answer 17: We appreciate the reviewer’s observation regarding the need for a more structured interpretation of the findings in the Discussion section and will take this into consideration in our revisions.

Comment 18:  Currently, the strengths and weaknesses of each PROM are not systematically compared. For instance, the statement that MAIA-2 is “most acceptable” is not adequately connected to specific psychometric data.

Answer 18: Thank you for your constructive comment. We have now highlighted the strengths and weaknesses of most relevant PROMs  and added a final section in the Discussion providing a direct comparison across all instruments. We believe this addition clarifies their relative psychometric performance and enhances the overall interpretability of our findings. It can be found on line 447 to 458.

Comment 19:  The authors are encouraged to discuss whether different PROMs assess distinct sub-constructs of body awareness (e.g., interoception, proprioception, embodiment), as this is essential for understanding construct heterogeneity.

Answer 19: Thank you for this valuable suggestion. We have added a new section in the Discussion section trying to summarize the discussed aspects previously in each PROM and to collect specific points about some sub-constructs assessed. It can be found on line 459 to 467.

Comment 20:  Practical recommendations are missing. Please consider adding suggestions for clinicians on which PROM(s) may be most appropriate for different contexts and guidance for researchers designing BA studies.

Answer 20: Thank you for this helpful suggestion. We have added practical recommendations for both clinicians and researchers. It can be found in the new section “Overall discussion about all the PROMs” (Lines 468-477)

Comment 21:  The Limitations section should be expanded to address heterogeneity of included populations, variability in translation procedures, inconsistent reporting across primary studies, and methodological limitations of relying solely on self-report measures.

Answer 21: Thank you for this valuable suggestion. We have expanded the Limitations section to address all the issues you raised, ensuring a more comprehensive discussion of the studies and instruments’ limitations. It can be found on line 483 to 494.

Comment 22:  The Conclusion is too brief and general. Please revise to highlight the main findings, practical implications, and future research directions.

Answer 22: Thank you for this helpful suggestion. We have revised the Conclusion section and we included clearer highlight about the main findings, practical implications, and directions for future research.

Kind regards,

The authors

This manuscript is a resubmission of an earlier submission. The following is a list of the peer review reports and author responses from that submission.

Round 1

Reviewer 1 Report

Comments and Suggestions for Authors

The manuscript “Patient-Reported Outcome Measures for Evaluating Body Awareness: A Systematic Review Using the COSMIN Methodology” presents a systematic synthesis of instruments designed to assess body awareness, with an emphasis on evaluating their psychometric properties and cross-cultural adaptation processes. Drawing on data from 47 studies, the authors review 12 distinct PROMs and more than 30 translations, identifying tools such as the MAIA-2 and BARQ-R as demonstrating comparatively stronger psychometric performance.

The introduction is well structured, clearly written, and grounded in the relevant literature, providing a coherent rationale for the study.

Several aspects, however, warrant further attention to strengthen the manuscript’s methodological rigor and completeness. The literature search was conducted in December 2024, more than eight months prior to manuscript submission. Methodological standards for systematic reviews recommend that searches be conducted no more than six months before submission to ensure currency of evidence. The authors are therefore strongly encouraged to refresh the search and update the results where applicable.

Given the stated focus on PROMs for body awareness, the absence of the Fremantle Awareness Questionnaire (including both regional and general versions) is notable. This instrument specifically targets body awareness, and its omission should be explicitly justified, or included in the study.

Although the supplementary file is comprehensive, it lacks essential methodological detail. Specifically, the criteria used to categorize studies with respect to cross-cultural translation process, structural validity, internal consistency, reliability, and measurement error are not reported. At present, only the criteria for hypotheses testing are described. The inclusion of explicit thresholds (e.g., Cronbach’s alpha values considered “good” or “very good,” ICC cut-offs for reliability) would improve transparency and reproducibility.

The manuscript would also benefit from integrating additional, recent literature to ensure completeness. In particular, the review by Oliveira (2022) on body mind relation (including body awareness) (https://pubmed.ncbi.nlm.nih.gov/35315163/) and the review of the Fremantle Awareness Questionnaire by Budzisz (2024) (https://www.sciencedirect.com/science/article/pii/S1526590024004504) should be incorporated into the discussion and contextualization of findings.

Addressing these points would enhance the manuscript’s methodological transparency, literature coverage, and scholarly contribution, ensuring it fully captures the intended scope and remains aligned with current best practices in systematic review methodology.

Reviewer 2 Report

Comments and Suggestions for Authors
Thank you for the opportunity to review an important topic. Here are some suggestions:
Abstract – The results section lacks coherence, as the authors first mention MAIA and MAIA2, then refer to MAIA-2 and BARQ-R. This could be clarified by combining into a single sentence. Additionally, the conclusions contradict the positive results obtained.
Introduction – Well structured. However, it relies heavily on outdated references (2003, 2009) throughout much of the section. It could be improved by incorporating more recent content, addressing PROMs in a general sense and their relationship with body awareness. There is a major error: according to the PROSPERO protocol, the authors refer to movement quality (also mentioned in the methodology), but this concept is only referenced once in the entire document and is not clearly defined.
Methodology – The values presented in Figure 1 are incorrect (see the screening and included categories). The rest of the methodology is appropriate, with a suitable assessment of study quality.
Discussion – Too brief given the results obtained. Lacks sufficient references (especially in section 4.1) to support analysis and discussion. Consider analyzing according to the PROMs mentioned in the abstract, or perhaps grouping the 10 questionnaires and conducting a structured analysis.
Conclusions – Ok.
References – Approximately 25% of the studies cited are over 10 years old.
Supplementary Material 3: Analysis of the Cross-Cultural Translation Process – This section is confusing, and it is unclear what the authors intend to convey. In the first column, it appears that some studies lack any PROM, but later it becomes evident that they are listed in order. It would be clearer to group by PROM in the first column or place the PROMs midway to show that multiple studies are associated with each one.